# The Thermodynamic Fingerprints of Ultra-Tight Nanobody–Antigen Binding Probed via Two-Color Single-Molecule Coincidence Detection

**DOI:** 10.3390/ijms242216379

**Published:** 2023-11-15

**Authors:** Benno Schedler, Olessya Yukhnovets, Lennart Lindner, Alida Meyer, Jörg Fitter

**Affiliations:** 1AG Biophysik, I. Physikalisches Institut (IA), RWTH Aachen University, 52074 Aachen, Germany; schedler@physik.rwth-aachen.de (B.S.); yukhnovets@physik.rwth-aachen.de (O.Y.); lennart.lindner@rwth-aachen.de (L.L.); meyer@physik.rwth-aachen.de (A.M.); 2ER-C-3 Structural Biology & IBI-6 Cellular Structural Biology, Forschungszentrum Jülich, 52425 Jülich, Germany

**Keywords:** single-molecule fluorescence detection, burst analysis, confocal fluorescence microscopy, brightness-gated two-color coincidence detection (BTCCD), antibody–antigen interaction, nanobodies, binding affinity, Van’t Hoff plot, thermodynamic parameters

## Abstract

Life on the molecular scale is based on a versatile interplay of biomolecules, a feature that is relevant for the formation of macromolecular complexes. Fluorescence-based two-color coincidence detection is widely used to characterize molecular binding and was recently improved by a brightness-gated version which gives more accurate results. We developed and established protocols which make use of coincidence detection to quantify binding fractions between interaction partners labeled with fluorescence dyes of different colors. Since the applied technique is intrinsically related to single-molecule detection, the concentration of diffusing molecules for confocal detection is typically in the low picomolar regime. This makes the approach a powerful tool for determining bi-molecular binding affinities, in terms of *K_D_* values, in this regime. We demonstrated the reliability of our approach by analyzing very strong nanobody-EGFP binding. By measuring the affinity at different temperatures, we were able to determine the thermodynamic parameters of the binding interaction. The results show that the ultra-tight binding is dominated by entropic contributions.

## 1. Introduction

Molecular interactions in living systems are relevant for almost all biological processes and lie at the heart of biology. The interactions between biomolecules are orchestrated in a highly tuned manner, closely related to their cellular functions. For a detailed understanding of the molecular mechanisms driving these processes, it is essential to investigate molecular interactions with regard to binding stoichiometry, specificity, affinity, and cooperativity [1,2]. In this respect, equilibrium constants for association between biomolecules represent a valuable measure of the binding affinity [3,4,5]. In particular, for high-affinity binding interactions, the quantitative evaluation of binding interactions is highly sensitive to many experimental conditions and to the performed procedures. Therefore, optimizing these conditions and procedures is critical for obtaining high-quality and reproducible data [5,6,7].

A large variety of experimental techniques are available to quantitatively evaluate the binding affinity between two biomolecules. In many of these methods, the target and the titrant molecules are mixed in different molar ratios to establish a binding equilibrium, in which the bound complexes and the unbound components are quantified. For such an approach, numerous methods have been developed, including calorimetric methods, optical methods, and gel filtration methods [2,4]. Nevertheless, the most commonly used methods are isothermal titration calorimetry (ITC) [8,9], microscale thermophoresis (MST) [10], surface plasmon resonance (SPR) [11], and several fluorescence-based methods, like fluorescence anisotropy (FA) [12], Förster resonance energy transfer (FRET) [13], or fluorescence correlation spectroscopy (FCS and FCCS) [14,15,16]. However, these methods are often reagent-limited and typically require nanomolar to millimolar concentrations of the involved reactants. For high-affinity interactions, typically with sub-nanomolar *K_D_* values, the quantities used for the corresponding measurements are simply too small to obtain reasonable binding curves due to detection sensitivity issues. Besides SPR and some electrophoretic mobility shift assays (EMSAs) [6], mainly fluorescence-based approaches [7,17,18] ensure the necessary measurement sensitivity. In particular, single-molecule fluorescence-based approaches are principally well suited for high-affinity binding studies [18,19,20]. However, many of these studies, as well as studies employing SPR, were performed with one immobilized binding partner, for which it is assumed that surface-immobilized binding partners can cause biased results and, therefore, are not ideal for studying in-solution interactions [21,22,23]. Thus, an approach which employs only diffusing binding partners would be beneficial for binding studies in solutions, including more physiological environments as crowded solutions. By attaching fluorophores of different colors to each of the binding partners, simultaneous dual-color fluorescence detection is the most straightforward to characterize binding interactions. For molecule concentrations in the nanomolar regime, this approach has been utilized to study molecular binding reactions with fluorescence cross-correlation spectroscopy (FCCS) [15,16,24]. For particularly strong binding interactions, one has to work with molecule concentrations in the picomolar regime, for which single-molecule counting has to be carried out. To perform binding studies in this regime, two-color coincidence detection (TCCD) [25,26,27] and the more recently developed brightness-gated two-color coincidence detection (BTCCD) [28,29] are effective approaches to quantify fractions of bound and unbound molecules in solution.

Herein, we developed and established protocols which enable us to make use of BTCCD to characterize high-affinity binding interactions. Since our approach allows us to measure both labeled binding partners in a sample solution molecule by molecule, we directly obtained reliable measures of the actual molecule concentrations in all samples and over extended incubation or observation times. In order to validate the reliability and the usefulness of our approach, we first characterized the hybridization of two complementary DNA strands, a well-established model system; see for example refs. [18,30]. In the next step, we characterized rather strong antigen–antibody binding, namely an EGFP and the related fluorescently labeled nanobody. In contrast to conventional antibodies, nanobodies consist of a single polypeptide chain (~14 kDa) but bind to their targets with similar affinities as full-length antibodies [31]. Here, we demonstrate the potential of the BTCCD approach for accurately measuring a strong non-covalent binding interaction, with corresponding *K_D_* values in the regime down to a few picomolar. Furthermore, measurements at different temperatures allowed the use of Van’t Hoff plots and the extraction of relevant thermodynamic parameters [32,33,34].

## 2. Results

### 2.1. Experimental and Methodical Design for the Determination of K_D_ Values Using BTCCD

As we will discuss below, the determination of reliable dissociation constants, in particular in the picomolar regime, is rather challenging and requires precautions and several cross-checks [3,4,5,6]. For bi-molecular binding, one assumes that one binding partner, the ligand *L*, can bind to another one, for example a protein *P*, to form a protein–ligand complex *P*·*L*. The reversible binding process
(1)$$P+L_{}{\overset{k_{a}}{\underset{k_{d}}{⇄}}}_{}P\cdot L$$
is characterized by the association rate *k_a_* and the dissociation rate *k_d_*. For equilibrium, the dissociation constant *K_D_* and the association constant *K_A_* are defined by
(2)$$K_{D}=\frac{k_{d}}{k_{a}}=\frac{1}{K_{A}}=\frac{{\left[P\right]}_{free}\cdot {\left[L\right]}_{free}}{\left[P\cdot L\right]}$$
where [*P*]*_free_*, [*L*]*_free_*, and [*P·L*] denote the concentration of unbound protein, unbound ligand, and bound protein–ligand complexes, respectively. The physical parameter that is usually employed to characterize the equilibrium is given by the binding fraction using the hyperbolic model
(3)$$f=\frac{\left[P\cdot L\right]}{{\left[P\right]}_{total}}=\frac{\left[P\cdot L\right]}{{\left[P\right]}_{free}+\left[P\cdot L\right]}=\frac{{\left[L\right]}_{free}}{{\left[L\right]}_{free}+K_{D}}$$

Our single-molecule approach, employing both binding partners labeled with differently colored dyes, allows the quantification of the concentration of all ingredients explicitly (i.e., [*P*]*_free_*, [*P*]*_total_*, [*L*]*_free_* [*L*]*_total_*, and [*P·L*]). If one assumes *L* is labeled with a red dye (where [*L*]*_total_* is, for example, calculated with *N_R_*; see Section 4.3), we obtain [*L*]*_free_* values by using one of the two possible formulas


[*L*]*_free_* = [*L*]*_total_* − *f_BR_*∙[*P*]*_total_*(4)


[*L*]*_free_* = [*L*]*_total_*·(1 − *f_RB_*)(5)
where the involved coincidence fractions *f_BR_* and *f_RB_* are defined and explained in Section 4.3. To determine *K_D_*, typically, the concentration of [*P*]_total_ is fixed to a certain value and [*L*] is varied with 0.1∙*K_D_* < [*L*] < 10∙*K_D_*. However, when measuring the binding curve, one has to choose the concentration of [*P*]*_total_* carefully. Only with [*P*]*_total_* << *K_D_* (i.e., a “binding” regime), one obtains a correct *K_D_* value, while with [*P*]*_total_* >> *K_D_* (i.e., a “titration” regime), reliable *K_D_* values are not determinable [5]. A transition between binding and titration regimes is called the intermediate regime. In this regime, the relation between the binding fraction, *K_D_*, and the protein and ligand concentrations can be described by a quadratic binding equation [3,5]
(6)$$f=\frac{\left({\left[P\right]}_{total}+{\left[L\right]}_{total}+K_{D}\right)-\sqrt{{\left({\left[P\right]}_{total}+{\left[L\right]}_{total}+K_{D}\right)}^{2}-4{\left[P\right]}_{total}\cdot {\left[L\right]}_{total}}}{2{\left[P\right]}_{total}}$$

In contrast to Equation (3), the quadratic model does not depend on the free ligand concentration but on the total concentrations of protein and ligand.

In order to make use of our single-molecule-based BTCCD approach (see Section 4.3), we developed a workflow specifying the sequence of individual steps in the sample preparation, in performing single-molecule measurements, and in data analysis (see Figure 1). We demonstrated all steps in an application by making use of a model system, namely the hybridization of two complementary DNA strands with a length of 24 base pairs (see Figure 1A).

Before a BTCCD-based binding study is truly considered for the potential binding pair, it should be either known or pre-tested that the expected *K_D_* value is in a regime suitable for our single-molecule approach. For the type of sample investigated here, we knew from previous studies that the affinity between both DNA strands is characterized by *K_D_* values in the picomolar regime [30]. Based on the fact that the distance between the dye label positions in the dsDNA molecule used is relatively large (see Figure 1A), FRET can only occur to a very small extent. Furthermore, we measure with pulsed interleaved excitation (see Section 4.2). As a consequence, possible energy transfer in the potential acceptor channel is not registered in the time window when we measure the direct excitation in this channel. Therefore, any possible influence of FRET on our results is negligible.

In a first straightforward approach, the strong inter-molecular binding of both strands was validated via an FCS analysis. As shown in Figure 1B, both strands exhibited an aggregation-free diffusion and a significant degree of inter-molecular binding between the complementary strands. The latter was demonstrated by the distinct dual-color cross-correlation amplitude, as expected for diffusing molecules in the nanomolar concentration regime. In order to make use of reliable single-molecule counting, we had to employ proper IPL threshold parameters for the initial burst selection (see Section 4.3). This was crucial for the extraction of meaningful *K_D_* values, since the determined molecule concentration of the binding partners in a sample is directly dependent on the number of initial bursts (see Section 4.3, Equation (8)). Therefore, the IPL threshold parameters should be chosen in a way that the signal and the background populations are well separated (see Section 4.3 for a definition of the relevant parameters). For samples labeled with bright fluorophores, this was easily achieved (Figure 1C), while for fluorescent proteins (like GFPs), this separation might be more challenging (see next subsection). Once the fluorescence detection of single molecules was established, we had to have ensured that the molecule concentration was constant over extended measuring times.

In particular, the evaporation of sample liquids and the unspecific binding of sample molecules to the surface of cover slide glasses are typically sources of problems in this respect. Proper surface-blocking procedures could significantly reduce these problems [29] and result in very stable molecule concentrations, even in the picomolar regime (Figure 1D). Based on the single-molecule counting approach, we made use of the recently developed brightness-gated two-color coincidence detection (BTCCD) [28,29] (see Section 4.3). Exemplarily, we obtained for a sample with a red-labeled DNA strand at ~5 pM (and with the blue-labeled strand at ~4 pM) fractions of coincidence bursts (i.e., fractions of bound dsDNA) in the order of 0.9 (Figure 1E). Both fractions (*f_RB_* and *f_BR_*) were rather similar and showed reliable (i.e., saturated) values at *n_br_* values of about 0.5 and 1, respectively.

The question of whether the obtained values could be used for a binding curve was answered by whether equilibrium for the binding reaction had been reached or not. In Figure 1F, an example is shown of how the binding reaction in the corresponding sample reached equilibrium. The kinetics of this reaction follow an exponential model, and here, after approximately five days, equilibrium was reached. Finally, the coincidence fractions, as measured for different ligand concentrations and always at equilibrium, were used for generating a binding curve (Figure 1G). A fit with the hyperbolic model (see Equation (3)) gives a *K_D_* value of about 0.24 pM. The obtained *K_D_* values indicate a surprisingly strong binding, with values well below 1 pM.

Unlike what was expected from Equation (3), the measured coincidence fractions *f_BR_* (e.g., in Figure 1G) did not reach a value of one at high ligand concentrations, but rather a value $f_{BR}^{max}$ < 1 (e.g., 0.6 in Figure 1G). Assuming the samples were fully equilibrated, $f_{BR}^{max}$ was only affected by the fraction of fluorescent ligand molecules. As previously stated, we used a fixed concentration of the blue-labeled strand [*B*] and varied the concentration of the red-labeled strand [*R*]. However, only fluorescent molecules contribute to these measured concentrations and the binding fraction. Thus, [*R*] and *f_BR_* need to be corrected by the factor $f_{BR}^{max}$; the latter can be directly obtained from the fit. The analogous correction factor $f_{BR}^{max}$ for [*B*] and *f_RB_* needs to be determined via an additional measurement (compare Figure 1, II.c). Taking those considerations into account, we obtained analytical mathematical expressions for a corrected version of the hyperbolic model for *f_BR_* and *K_D_* values, which are described in Section 4.4. Unlike the hyperbolic model, the quadratic model cannot be corrected for $f_{BR}^{max}$ in an analytical way. Thus, the correction of [*R*] needs to be performed in an iterative manner. A comparison of corrected and uncorrected data from the binding interactions of complementary DNA strands is shown in Appendix A. In general, the quality of the fit for corrected and uncorrected data is rather similar but with partly different *K_D_* values.

Although, in principle, we obtained reasonable titration curves for the extreme strong binding affinity between both DNA strands, we cannot fully trust the obtained *K_D_* values since the data were measured in the titration regime. Furthermore, numerous measurement points of the corresponding titration curve were obtained from samples with ligand concentrations below an assumed minimum value (the smallest measurable concentration was approx. 0.1–0.3 pM, depending on the dye used). In this sense, the presented result shows that, at or below the limit of applicability of our method, only a rough guide value for the *K_D_* values can be provided. Since the binding strength between DNA strands depends on the length of the complementary strands [30,35], a shorter length of chosen strands would most probably lead to weaker binding affinity and would most likely ensure our method had better applicability. However, here, we do not want to focus any further on DNA hybridization, but we will expand and validate our approach by analyzing another high-affinity binding system, as demonstrated in the next section.

### 2.2. Measuring the Nanobody-EGFP Binding Affinity and Thermodynamic Analysis

In order to elucidate details of the high-affinity antibody–antigen binding interaction, we employed a red-labeled nanobody (the so-called GFP-booster) and a purified recombinant enhanced green fluorescent protein (EGFP); see Figure 2A. A special situation for our method is given by the fact that one binding partner, the EGFP, is already intrinsically fluorescent. On the one hand, this is advantageous because no labeling step with an organic fluorophore is required. On the other hand, EGFP fluorescence at the single-molecule level suffers from lower molecular brightness and more non-fluorescent molecules (due to incomplete EGFP maturation or more pronounced photo-destruction) [28,36] as compared to the Atto647N-labeled nanobody. However, a study with this system can provide evidence as to whether GFPs are applicable or not. In order to investigate the robustness and reliability of our method, we performed binding reactions at various temperatures. Thereby, weaker binding (i.e., a larger *K_D_* value) was expected for the reaction at the higher temperature as compared to that performed at the lower temperature.

After ensuring the samples were fully equilibrated, coincidence fractions were measured with free ligand (nanobody) concentrations [*R*]*_free_* between 0.05 and 120 pM. The fitting results with corrected [*R*]*_free_* and *K_D_* values are shown for three different temperatures in Figure 2.

For all measured temperatures, the corresponding equilibrium curves (Appendix A) and titration curves with fits for uncorrected and corrected data (corrected [*R*]*_free_* and *K_D_* values; see Section 4.4) are given in Appendix A, respectively. The obtained *K_D_* values are in the regime of a few picomolar. These values indicate much stronger binding affinity in the complex, as observed for another nanobody-GFP complex that was described earlier. In this similar but not identical system, *K_D_* values between 0.3 and 1.4 nM were measured [37]. However, the analysis of our data revealed that we observed shorter equilibrium times and larger *K_D_* values (i.e., weaker binding) for samples at higher temperatures as compared to those at lower temperatures. Since we measured titration curves at a total of eleven different temperatures between 20 and 45 °C, we made use of Van’t Hoff plots (see Section 4.5, Equation (18)) and, thereby, have access to the thermodynamic parameters ∆*H*, ∆*S*, ∆*G*, and ∆*C_P_* of the binding reaction (see Figure 3).

As can be observed in almost all thermodynamic binding studies, our Van’t Hoff plots also show an upward curvature, indicating a negative value for the change in heat capacity ∆*C_P_* [32,38]. Since hydration water (which partly interacts with the hydrophobic surface) and bulk water have different properties, the change in heat capacity due to the release of water molecules upon complex formation is proportional to the amount of surface area involved. In this respect, a negative ∆*C_P_* value indicates the burial of non-polar or hydrophobic surface areas upon binding [32,33].

As shown in Table 1, the nanobody-EGFP interaction has a strikingly strong entropic contribution that stabilizes the binding of the complex. This observed mechanism is remarkable because most high-affinity interactions, as well as most antibody–antigen bindings, are enthalpy-driven; see complexes (3)–(6) in Table 1 and ref. [38]. However, for nanobody–antigen interactions [34], as well as for other high-affinity bindings [39,40], recognition mechanisms were already found that showed a strong entropic contribution to high binding affinities. Especially in the case of nanobody–antigen binding, approaches for optimizing binding affinities are of special interest. On the one hand, binding entropy, which is highly dependent on the hydrophobic effect, is assumed to be easier to optimize than binding enthalpy. It has been shown that increasing the hydrophobicity of a compound is a straightforward way of increasing its binding affinity. On the other hand, an increase in hydrophobicity can decrease the target selectivity and cause unfavorable effects on the compound’s solubility [41].

## 3. Discussion

The presented BTCCD approach is based on single-molecule counting and, therefore, gives highly direct access to characterize the ensemble of molecules in a sample. In order to achieve this high detection sensitivity, both binding partners need to be labeled with fluorescent probes. Fortunately, the degree of labeling with suitable fluorophores does not need to be 100% in order to obtain reliable results. The reason for sub-optimal fluorescent binding partners is often incomplete labeling, partly by photo-destructed fluorophores, or a certain number of fluorophores not emitting a sufficient number of photons to be recognized as a burst (i.e., reduced molecular brightness). However, as demonstrated, this can be handled as long as the label ratio is at least ~40%. Below this threshold, errors become too large and the obtained *K_D_* values are not determined sufficiently precisely. Such sub-optimal fluorescently labeled binding partners (for example, the EGFP) can nevertheless contribute successfully to proper binding assays, partly because complementary experimental parameters (like *f_RB_* and *f_BR_*) can help to circumvent shortcomings in the employed samples. Furthermore, complementary experimental parameters can help to cross-validate the obtained *K_D_* values (see Appendix A). However, the presented approach is generally restricted by essentially two limits: (i) the low concentration limit, which does not allow measuring concentrations below ~0.1 pM due to the very small number of bursts (i.e., diffusing molecules), which causes insufficient counting statistics or extremely long measuring times; (ii) the high concentration limit (above ~100 pM), above which one would typically no longer observe single molecules in the confocal detection volume. The latter causes chance coincidence events, which strongly distort the determined *f_RB_* and *f_BR_* values; see, for example, ref. [29]. As a consequence, this concentration regime would allow measuring *K_D_* values between 0.5 and 10 pM. Another potential problem in our approach could be the scenario where the bound dyes have a different brightness between the bound and unbound states. Fortunately, the single-molecule approach of our method enables us to detect such behavior in our data. In such a case, it would be best to prepare a sample in which the dye is bound to a different site on the protein. This sample must then be used to validate that the aforementioned problem does not occur anymore. In the case of using GFPs in a binding assay, in addition to lower molecular brightness in the fluorescent molecules, a pronounced temperature dependence of the brightness can play a role. This can potentially lead to falsified results if the *K_D_* values are determined as a function of temperature (for example, in Van’t Hoff plots). An effective way to circumvent such problems is not to vary the GFP-based binding partner in the molecular concentration when measuring titration curves but to vary the molecule concentration of the other dye-labelled binding partner. Since there is no comparable temperature dependence with the other fluorescent dyes, we can obtain trustful results, as we demonstrated in the case of EGFP-nanobody binding.

Further challenges for measuring extremely high binding affinities are not related to aspects of single-molecule fluorescence detection but are associated with common difficulties. On the one hand, related studies may suffer from the unspecific binding of molecules to surfaces of test tubes or cover slides. This can significantly alter the targeted concentration of freely diffusing molecules that are required for many binding assays. On the other hand, high-affinity binding is often linked to rather long incubation times required to reach equilibrium; see refs. [3,4,5]. In order to judge the reliability of our methodical approach, a comparison of our results with those from other complementary standard techniques would be helpful. So far, we have already observed reasonable agreement between the binding affinity of nanobody-EGFP binding, which was characterized by a *K_D_* ~ 2 pM at room temperature, as measured with another label-free method [17] (personal communication ChromoTek, Planegg, Germany), and our results.

As demonstrated in this work, the BTCCD approach is, on the one hand, a very reliable method to analyze high-affinity binding, but on the other hand, it is also limited to the picomolar (*K_D_*) binding regime. However, besides the fact that it is technically challenging to obtain trustworthy *K_D_* values in this regime, several high-affinity molecular interactions of great biological significance can only be investigated in this regime. In addition to DNA hybridization [18,30,35] (also relevant for the amplification of DNA sequences or DNA sequencing) and antigen–antibody binding [7,42], which were studied in this work, also DNA-protein binding [6,43], protein–inhibitor binding or toxin–antitoxin [44] binding can exhibit extremely high binding affinities and are, therefore, further potential targets for the BTCCD approach. Only recently was it reported that intrinsically disordered proteins (IDPs) can also exhibit surprisingly ultra-tight binding to their target molecules, with *K_D_* values of a few pM [39,45,46]. Here, it would be of interest to better understand the differences between the binding of IDPs and globular proteins to their targets, a goal that is well within the scope of the presented BTCCD approach.

## 4. Materials and Methods

### 4.1. Sample Preparation

For reversible DNA binding (hybridization) studies, we employed the following dye-labeled DNA strands: 5′-Atto647N GGC GAT CTC TGT TTA CAA CTC CGA-3′ and 5′-Alexa488 TCG GAG TTG TAA ACA GAG ATC GCC-3′ (IBA, Göttingen, Germany). If not stated otherwise, the DNA samples were measured in PBS buffer at pH 7.4. The manufacturer of the labelled DNA strands specified a label ratio of approx. 90%. In practice, we also received samples with lower label ratios from the manufacturer, partly causing maximal fractional binding ratios of *f_max_* ~ 60% in samples for titration curve measurements. For antibody–antigen binding studies, an anti-GFP VHH/nanobody (Alpaca single-domain antibody, monovalent VHH binding) conjugated with Atto647N (degree of labeling: 2 fluorophores per nanobody) and EGFP (both from ChromoTek, Planegg, Germany) were used. Samples were aliquoted and measured in PBS buffer (14 mM KH_2_PO_4_, 36 mM K_2_HPO_4_, 150 mM NaCl, pH 7.2) and stored at 4 °C. All buffers contained 0.1% NaN_3_ and 0.005% Tween 20. For binding reaction equilibration, samples were stored in a TS-100C thermo shaker (SIA Biosan, Riga, Latvia) at the required temperature.

### 4.2. Confocal Microscopy and Data Acquisition

Confocal measurements were performed using a MicroTime200 (PicoQuant, Berlin, Germany). The fluorophores were excited using LDH-D-C 485B and LDH-D-C 640B lasers with 485 nm and 640 nm emission (PicoQuant, Berlin, Germany) and a power in the regime of a few 10 μW. For BTCCD measurements, lasers were operated in a pulsed interleaved excitation (PIE) scheme, in which blue and red excitation are alternated in order to directly excite both channels [47]. The excitation light was focused and collected using a high-numerical-aperture water immersion objective (UPLSAPO 60x; Olympus, Hamburg, Germany) and directed through a 75 µm pinhole. The emission signal was separated using a dichroic mirror (T600lpxr, Chroma Technology, Olching, Germany) and filtered with band pass filters of 535 nm (FF01-535/55-25, Semrock, Rochester, NY, USA) and 685 nm (ET685/80m, Chroma Technology, Olching, Germany) for the blue and the red channels, respectively. Photons were detected using single-photon avalanche diodes (τ-SPAD, PicoQuant, Berlin, Germany; COUNT-T, Laser Components, Olching, Germany). Sample temperatures were adjusted via the use of a sample chamber P-Set 2000 and an objective ring (both from PeCon, Erbach, Germany) both connected to a temperature-controlled water bath circulator (DYNEO DD-300F refrigerator/heating circulator, Julabo, Seelbach, Germany). In order to avoid the extensive evaporation of water immersion liquid during long-lasting measurements above room temperature, we limited the time of individual measurements and used renewed samples. All samples were measured on PEGylated cover slides [29]. The typical time for measuring the data of a sample with specific concentrations of the related binding partners (i.e., one point in the titration curve) was about two hours. The concentration of differently labeled molecules in the samples was first determined with FCS. After diluting to the target single-molecule concentrations (i.e., pM concentration regime), the real single-molecule concentration was determined with a burst analysis (see below).

### 4.3. Burst Analysis and BTCCD

In order to identify and count individual (i.e., single) molecules, the inter photon lag (IPL) trace was calculated from acquired intensity traces [48] and a corresponding burst analysis was performed, as already described in detail previously [29]. Briefly, single bursts in the red and the blue detection channels were discriminated from the background by applying a suitable threshold (usually ~100–200 μs); see Figure 1C. Typical data sets contain a number of 10^3^–10^4^ accepted bursts. In order to calculate the molar concentrations *C* = *N*/(*N_A_*·*V_eff_*) of the labeled molecules in the sample (in the single-molecule regime), the average number of detected molecules *N* and the dimension of confocal detection volume need to be known (for each color). *N* can be calculated from the total number of detected bursts, *B_meas_*, and the dwell time of diffusing molecules visible in the detected bursts, *τ_d_*, by considering the total fluorescence time *t_F_* [49]. The total fluorescence time is defined as the product of the total measurement time *t_meas_* and the probability of detecting a molecule at a given time (1 − exp(−*N*)) or through the product of *B_meas_* and <*τ_d_* > with
(7)$$t_{F}=\left[1-\exp\left(-N\right)\right]\cdot t_{meas}=B_{meas}\cdot \langle \tau_{d}\rangle$$

This equation can then be used to determine the average number of detected molecules with
(8)$$N=-\ln\left(1-B_{meas}\frac{\langle \tau_{d}\rangle }{t_{meas}}\right)$$

In order to determine trustable fractions of bound molecules, an improved version of the conventional two-color coincidence detection (TCCD; see for example [25]) was employed. Here, the so-called brightness-gated two-color coincidence detection (BTCCD) overcomes the problem of coincidence fraction underestimation, caused by incomplete detection volume overlap for different excitation wavelengths and lens aberrations. In order to estimate the coincidence fraction precisely, only molecules that diffused through both confocal detection volumes should be considered for analysis. As a consequence of incomplete detection volume overlap for the two wavelengths, it was assumed that molecule trajectories which correspond to bright bursts with a high number of emitted photons are more likely to touch both volumes [28]. In contrast, molecule trajectories corresponding to dim bursts with only a small number of emitted photons were more likely to touch one of the volumes only slightly. For each accepted burst, the burst intensity, i.e., the number of photons detected between the start and end time, and the mean number of photons per burst were calculated. To perform a coincidence analysis, the brightness threshold *n_br_*, defined as the number of photons in a burst, normalized to the mean number of photons, was continuously increased. The coincidence was calculated for the red channel (*f_RB_*) and the blue channel (*f_BR_*) independently with
(9)$$f_{RB}(n_{br})=\frac{N_{RB}(n_{br})}{N_{R}(n_{br})},\;\;\;f_{BR}(n_{br})=\frac{N_{BR}(n_{br})}{N_{B}(n_{br})}$$
where *N_RB_* and *N_BR_* are the number of coincident bursts in the red and blue channels, and *N_R_* and *N_B_* are the total number of selected red and blue bursts, respectively. For each value of the brightness threshold, only bursts that had more photons as defined by the brightness threshold were considered for analysis. Two bursts were considered as coincident if the start or end time tag of one burst was within the start and end time tags of the other burst. Coincidence fractions increase with the increase in *n_br_* and eventually saturates once all bursts considered for the analysis correspond to molecule trajectories through both volumes (see Figure 1E). The probability of having more than one molecule in the detection volume at the same time is known as chance coincidence and has to be considered for samples with a concentration higher than a few tens of picomolar. Therefore, the coincidence fractions have to be corrected as described earlier [29]. A complete description of the BTCCD method can be found in refs. [28,29].

### 4.4. Corrections for Binding Fractions and Resulting K_D_ Values

Although we expected from Equation (3) that the measured coincidence fractions *f_BR_* would reach a value of one at high ligand concentrations, we obtained values of $f_{BR}^{max}$ < 1. Since only fluorescent molecules contribute to these measured concentrations and the binding fraction, [*R*] and *f_BR_* need to be corrected with the factor $f_{BR}^{max}$. The latter can be directly obtained from the performed fit. The analogous correction factor $f_{BR}^{max}$ for [*B*] and *f_RB_* need to be determined via an additional measurement (see Figure 1, II.c). Taking those considerations into account, we obtained an analytical mathematical expression for a corrected version of the hyperbolic model
(10)$$f_{BR}=f_{BR}^{max}\cdot \frac{{\left[R\right]}_{free}}{{\left[R\right]}_{free}+K_{D}^{app}}\;\mathrm{with}\;{\left[R\right]}_{free}={\left[R\right]}_{tot}-f_{BR}\cdot \frac{{\left[B\right]}_{tot}}{f_{RB}^{max}}$$
or
(11)$${\left[R\right]}_{free}={\left[R\right]}_{tot}\cdot \left(1-\frac{f_{RB}}{f_{RB}^{max}}\right)$$
with a final value of
(12)$$K_{D}=\frac{K_{D}^{app}}{f_{BR}^{max}}$$

Unlike the hyperbolic model, the quadratic model cannot be corrected for $f_{BR}^{max}$ in an analytical way. Thus, the correction of [*R*] needs to be performed in an iterative manner:(13)$$f_{BR}=f_{BR}^{max}{}_{i}\frac{\left(\left[B\right]+{\left[R\right]}_{i}+K_{D}\right)-\sqrt{{\left(\left[B\right]+{\left[R\right]}_{i}+K_{D}\right)}^{2}-4\left[B\right]\cdot {\left[R\right]}_{i}}}{2\left[B\right]}\;\mathrm{with}\;{\left[R\right]}_{i+1}=\frac{\left[R\right]}{f_{BR}^{max}{}_{i}}\;\mathrm{and}\;\left[B\right]=\frac{{\left[B\right]}_{meas}}{f_{RB}^{max}}$$
where *i* gives the number of iteration steps (for details, see Appendix A). Typically, three iteration steps were sufficient to obtain convergence in the results. In both models, $f_{BR}^{max}$ and *K_D_* were free fitting parameters. The impact of these corrections was investigated via the binding of the complementary DNA strands. First, we performed fits with uncorrected data points; i.e., [*R*]*_free_* and $K_{D}^{app}$ values were not explicitly corrected (i.e., we used Equations (3)–(5) but not Equations (10)–(12)). The corresponding hyperbolic and quadratic fits for DNA data are shown in Appendix A. Furthermore, we observed a deviation in the two *K_D_* values for both hyperbolic fits (0.24 and 0.09 pM; see Appendix A), which was unexpectedly large. In the next step, we were able to obtain reasonable corrected [*R*]*_free_* values for using *f_RB_* (see Equation (11) and Appendix A, but not when using *f_BR_* (see Equation (10)). In the latter case, we obtained very small [*R*]*_free_* values (partly < 0.01 pM), which excluded these data from a reasonable analysis (therefore, a graph is not shown). In the case of quadratic fits, a correction corresponding to Equation (13) was possible again (Appendix A). In general, the quality of the fit for the corrected and uncorrected data was rather similar. In contrast to the stronger deviation between the obtained *K_D_* values from the uncorrected data, the corrected data (including corrected *K_D_* values according to Equation (12)) exhibited more matching results (however, one data set was missing). A much more consistent picture emerged for the nanobody-EGFP data. Here, at higher temperatures, the corrected data (compared to the uncorrected data) at least showed slightly higher *K_D_* values (see Appendix A).

### 4.5. Thermodynamics of the Bi-Molecular Binding

For bi-molecular binding, the free-energy change between the unbound and bound state is called binding free energy. It is known as
(14)$$\Delta G_{bind}^{}=RT\cdot \ln\left(K_{D}\right)$$
where *R* = 8.314 J∙(mol·K)^−1^ is the gas constant. The temperature dependence of the dissociation constant is given by the Van’t Hoff equation:(15)$$\begin{array}{l} \Delta G_{bind}^{}=\Delta H_{bind}^{}-T\Delta S_{bind}^{}\\ \Leftrightarrow \ln\left(K_{D}\right)=\frac{\Delta H_{bind}^{}}{RT}-\frac{\Delta S_{bind}^{}}{R}\\ \Leftrightarrow K_{D}=\exp\left(\frac{\Delta H_{bind}^{}}{RT}\right)\cdot \exp\left(\frac{-\Delta S_{bind}^{}}{R}\right) \end{array}$$

Here, ∆*H_bind_* and ∆*S_bind_* denote the enthalpic and the entropic parts of the binding free energy ∆*G_bind_*, respectively. The temperature dependence of ∆*H*(*T*) and ∆S(*T*) is directly related to the change in the systemic heat capacity (at constant pressure) by [9,32]
(16)$$\Delta C_{P}={\left(\frac{\partial \Delta H}{\partial T}\right)}_{P}=T{\left(\frac{\partial \Delta S}{\partial T}\right)}_{P}$$

The integration of Equation (16) with the assumption of a constant heat capacity change upon ligand binding leads to the so-called integrated Van’t Hoff equation with
(17)$$\begin{array}{l} \Delta H(T)=\Delta H_{T0}^{}+\Delta C_{P}(T-T_{0})\\ \Delta S(T)=\Delta S_{T0}^{}+\Delta C_{P}\ln\left(\frac{T}{T_{0}}\right) \end{array}$$
which gives the enthalpy and entropy changes at a particular temperature *T* in terms of entropy, enthalpy, and heat capacity changes at a second conveniently chosen reference temperature *T*_0_ [32,33]. Inserting Equation (17) in Equation (15) leads to the temperature dependence of *K_D_* with
(18)$$\ln\left(K_{D}\right)=\frac{\Delta H_{T0}^{0}}{RT}-\frac{\Delta S_{T0}^{0}}{R}-\frac{\Delta C_{P}}{R}\left[\left(\frac{T_{0}}{T}-1\right)-\ln\left(\frac{T_{0}}{T}\right)\right]$$

## Data Availability

The data and software that are contained within the article are available on request from the corresponding author.

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
