# Peer review of "The Thermodynamic Fingerprints of Ultra-Tight Nanobody–Antigen Binding Probed via Two-Color Single-Molecule Coincidence Detection"

_ijms, 2023, doi:10.3390/ijms242216379_

Round 1

Reviewer 1 Report

Comments and Suggestions for Authors

The manuscript by Schedler et al. used the brightness-gated two-color coincidence detection to measure the binding constants and the thermodynamics of a nanobody-antigen binding. The authors presented the workflow and tested it first with the hybridization of complementary DNA strands before employing the approach to nanobody-EGFP binding. The authors did an excellent job in providing the necessary mathematical foundations behind the method and the data analysis, as well as the caveats and necessary mathematical corrections. They also were able to demonstrate the feasibility and reliability of the approach. While this method is highly reliable, its application is limited to very strong bimolecular interactions in the picomolar range (around 0.5 – 10 pM). Overall, the manuscript is informative and the data analysis was thorough.

Author Response

The manuscript by Schedler et al. used the brightness-gated two-color coincidence detection to measure the binding constants and the thermodynamics of a nanobody-antigen binding. The authors presented the workflow and tested it first with the hybridization of complementary DNA strands before employing the approach to nanobody-EGFP binding. The authors did an excellent job in providing the necessary mathematical foundations behind the method and the data analysis, as well as the caveats and necessary mathematical corrections. They also were able to demonstrate the feasibility and reliability of the approach. While this method is highly reliable, its application is limited to very strong bimolecular interactions in the picomolar range (around 0.5 – 10 pM). Overall, the manuscript is informative and the data analysis was thorough.

Author response: We would like to thank this reviewer for the very positive vote on the submitted manuscript.  

Reviewer 2 Report

Comments and Suggestions for Authors

The paper describes the investigation of ligand binding processes with small dissociation constant, using brightness gated two-color coincidence detection method.

Minor comments:

Line 110 and 111: please define f_BR and F_RB!

Fig 1 Letter size is too small.

Scientific question: Should the FRET taken into account in case of hybridization experiments? Can it influence the results? This should be explained.

The paper is suitable for publication after the above suggested changes.

Author Response

Minor comments:

Line 110 and 111: please define f_BR and F_RB!

Author response: We referred the required definitions for fBR and fRB in the manuscript, see page 3.

Fig 1 Letter size is too small.

Author response: We increased the letter size in Figure 1.  

Scientific question: Should the FRET taken into account in case of hybridization experiments? Can it influence the results? This should be explained.

Author response: Because of the distances between the dye label positions in the dsDNA molecule used, FRET can only occur to a very small extent (~ 5%). Moreover, since we measure with pulsed interleaved excitation, a possible energy transfer in the potential acceptor channel (red) is not measured in the time window when we measure the direct excitation in this (typically red) channel. Thus, even with possible energy transfer, there is no appreciable influence on the results. We would like to thank the reviewer for this advice and explained the corresponding facts in a short comment in Section 2.1 (page 3)

Reviewer 3 Report

Comments and Suggestions for Authors

Schedler et al. report an application of two-colour single-molecule coincidence detection to measure the binding characteristics of high-affinity biomacromolecular complexes. The authors detail a workflow for estimation of the equilibrium dissociation constant, examining two systems: a DNA hybridisation equilibrium and a nanobody interaction with green fluorescent protein. This work provides proof-of-principle validation for the method as suitable for characterising these kinds of high-affinity interaction.

The manuscript overall is written clearly and logically, and the data generally support the conclusions. Nevertheless, there are several aspects, in my view, that could benefit from revision/clarification, including for the benefit of a broader audience.

1. The DNA hybridisation experiments yield in part a negative result – the authors state they “cannot fully trust” the measured dissociation constant(s). Thus, the applicability of the method is not entirely clear. If the goal is to report a widely-usable workflow for this technique, I would encourage the authors to provide additional data from optimised experimental conditions that demonstrate reliable KD estimation. At a minimum, in my view, there should be a more thorough elaboration of steps that could be taken to resolve the issue, beyond the relatively generic statements on lines 213-220.

2. The authors report that the fractional binding reached a maximum of 60% after the DNA-based system equilibrated. If I am understanding correctly, they are attributing the remaining 40% to unlabeled (or photobleached?) complementary strand which competes for the colocalisation signal. 60% labeling seems low for oligonucleotides of this type, especially if they are fully synthetic. The authors should provide a more detailed explanation for this effect, and also quantitation data for the labeling efficiency.

3. Related to point 2, conjugation of dyes to nucleic acids can influence both the photophysical properties of the dyes and potentially the hybridisation properties of the nucleic acids. The effects on dye photophysical properties can be distinct between single-stranded and double-stranded DNA, which would seem relevant to the threshold-based fluorescence measurement here. Of course, any potential effects could be favourable, since attachment to nucleic acids often enhances brightness. But, in my view, the authors should discuss these issues for the specific dyes in question, and the potential impact on their results.

4. The nanobody-GFP results appear to be more robust, but again the authors report a surprising ~1,000-fold greater affinity relative to previous measurements of a similar system. The explanation offered by the authors needs to be more clear and explicit – they seem to be suggesting that differences in experimental conditions are responsible, but a direct comparison to experimental conditions in the past study is not made, beyond a reference (number 37). This should be added, with a clear statement of how the differing conditions could cause this very significant difference in affinity.

5. The fluorescence of GFP can be markedly temperature-sensitive (e.g., Savchuk et al., Sci. Rep. 9(7535)). The authors should explicitly state, in my view, whether/how this was taken into account in the variable-temperature analysis used to construct the Van’t Hoff plots.

Minor:

1. The acronym “IPL” is not defined in the text, as far as I could see; the authors should include the definition for the benefit of a broader audience.

2. The authors mention an iterative approach to correct the quadratic binding isotherm, but I think it would be helpful to explain more clearly why this is a valid approach — again, for the benefit of a broader audience who may not have encountered it before.

Author Response

Schedler et al. report an application of two-colour single-molecule coincidence detection to measure the binding characteristics of high-affinity biomacromolecular complexes. The authors detail a workflow for estimation of the equilibrium dissociation constant, examining two systems: a DNA hybridisation equilibrium and a nanobody interaction with green fluorescent protein. This work provides proof-of-principle validation for the method as suitable for characterising these kinds of high-affinity interaction.

The manuscript overall is written clearly and logically, and the data generally support the conclusions. Nevertheless, there are several aspects, in my view, that could benefit from revision/clarification, including for the benefit of a broader audience.

  1. The DNA hybridisation experiments yield in part a negative result – the authors state they “cannot fully trust” the measured dissociation constant(s). Thus, the applicability of the method is not entirely clear. If the goal is to report a widely-usable workflow for this technique, I would encourage the authors to provide additional data from optimised experimental conditions that demonstrate reliable KD estimation. At a minimum, in my view, there should be a more thorough elaboration of steps that could be taken to resolve the issue, beyond the relatively generic statements on lines 213-220.

Author response: The fact that the results of DNA hybridization cannot be fully trusted is not primarily a negative result in the strictest sense of the word, but confirms the discussed limitations of the applications. These limitations are mainly determined by the smallest measurable sample concentrations (limit value approx. 0.1- 0.3 pM, depending on the dye). In the case discussed here, the sample shows a KD value of about 0.24 pM based on the measurements.  (i) Numerous measurement points of the corresponding titration curve show significantly smaller concentrations for the ligand than the limit value mentioned above. (ii) the selected concentration for the other binding partner of 1 pM is significantly higher than the KD value and therefore no longer in the binding regime. Both facts demonstrate that these measurements do not meet the requirements for a fully trustworthy measurement in the sense of our approach. In this sense, we also cannot simply optimize the experimental conditions, which would require a sample with a weaker binding affinity (e.g., a new sample with shorter chain length, which is beyond the scope of the presented work). We made this point clearer and extended the discussion in the revised manuscript on page 6.  

  1. The authors report that the fractional binding reached a maximum of 60% after the DNA-based system equilibrated. If I am understanding correctly, they are attributing the remaining 40% to unlabeled (or photobleached?) complementary strand which competes for the colocalisation signal. 60% labeling seems low for oligonucleotides of this type, especially if they are fully synthetic. The authors should provide a more detailed explanation for this effect, and also quantitation data for the labeling efficiency.

Author response: We agree with the reviewer that the obtained value for fractional binding of the dsDNA pair with fmax ~ 60 % is quite low. In numerous other measurements we have achieved also higher values, of even up to 80-90 %. The manufacturer specified label ratios of approx. 90 %. However, in practice we have also received samples with lower label ratios from the manufacturer. In addition, the measurable double label ratio also suffers from photobleaching or similar effects due to longer storage times (e.g. for more than a few months). Importantly, as we explained in detail in Section 4.4 and demonstrated with samples in Section 2.2, samples with fmax values lower than 80 or 90 % can be employed as well for trustful determination of KD values. We added the required explanation and information in the revised version of the manuscript (Section 4.1, page 10).           

  1. Related to point 2, conjugation of dyes to nucleic acids can influence both the photophysical properties of the dyes and potentially the hybridisation properties of the nucleic acids. The effects on dye photophysical properties can be distinct between single-stranded and double-stranded DNA, which would seem relevant to the threshold-based fluorescence measurement here. Of course, any potential effects could be favourable, since attachment to nucleic acids often enhances brightness. But, in my view, the authors should discuss these issues for the specific dyes in question, and the potential impact on their results.

Author response: In our measurements (with Alexa 488 and Atto647 labelled DNA strands) we did not observe any significant changes in the molecular brightness of the dyes between bound and unbound states. Beyond the DNA samples, a binding-induced change in molecular brightness can of course be a problem (in terms of threshold-based fluorescence measurements). And of course this could potentially also occur in the case of dye-labelled proteins (e.g. due to local quenching). However, no such behavior was observed in our measurement, even in the case of the protein-based case (EGFP booster). Nevertheless, we agree with the reviewer that this issue is of general importance and must be considered with regard to the applicability of our approach We have therefore added a few sentences in the Discussion (Scheme 1, point IIg, Section 3, page 9). 

  1. The nanobody-GFP results appear to be more robust, but again the authors report a surprising ~1,000-fold greater affinity relative to previous measurements of a similar system. The explanation offered by the authors needs to be more clear and explicit – they seem to be suggesting that differences in experimental conditions are responsible, but a direct comparison to experimental conditions in the past study is not made, beyond a reference (number 37). This should be added, with a clear statement of how the differing conditions could cause this very significant difference in affinity.

Author response: A closer look at the original literature revealed experimental KD values of 0.3 - 1.4 nM for the cited GFP-GFP-nanobody complex (see ref. 37). The cited system (GFP and nanobody optimized in llama cells) does not completely match the system used in the study (EGFP and nanobody optimized in alpaca cells). In addition, it is not known from the literature whether different approaches and cycles were used to optimize the binding affinity of the nanobody to the target. Thus, the significant difference (~factor 100) in the reported binding affinity must be seen against this background. We mentioned these facts shortly in the revised version of the manuscript (see page 8).   

  1. The fluorescence of GFP can be markedly temperature-sensitive (e.g., Savchuk et al., Sci. Rep. 9(7535)). The authors should explicitly state, in my view, whether/how this was taken into account in the variable-temperature analysis used to construct the Van’t Hoff plots.

Author response: We thank the reviewer for this important hint. We indeed observe a notable decrease in the molecular brightness of EGFP with increasing temperature, for example in the range of 20° to 40°C, similar to that mentioned in the literature. In principle, such a temperature-dependent change in molecular brightness has the potential to distort the temperature dependence of the KD values. However, this would only occur if this temperature dependent brightness occurs with the binding partner whose molecular concentration is varied for the measurement of the titration curve (i.e. ligand). In our case, however, the concentration of the dye-labelled nanobody is varied. When determining the coincidence fBR, only a slightly different sized sample population is used to determine fBR if the measured concentration of EGFP changes (due to a brightness-dependent shift in the threshold). This has no (or negligibly small) influence on the KD value determined at a specific temperature. Therefore, there is no problem with this effect in our approach and we can trust the obtained Van’t Hoff plots. Importantly, there is no comparable temperature dependence with the other fluorescent dyes used (according to the manufacturer’s specification and based on our measurements). Importantly, there is no comparable temperature dependence with the other fluorescent dyes used. Nevertheless, in cases where GFPs are used in such binding assays, there is the important limitation mentioned above. Therefore, these facts are also mentioned in the discussion (page 10) of the revised version.     

Minor:

  1. The acronym “IPL” is not defined in the text, as far as I could see; the authors should include the definition for the benefit of a broader audience.

Author response: We referred now (see page 4) to the definition (page 11).

  1. The authors mention an iterative approach to correct the quadratic binding isotherm, but I think it would be helpful to explain more clearly why this is a valid approach — again, for the benefit of a broader audience who may not have encountered it before.

Author response: The details explaining this approach are now given in the Supplementary Materials, see Scheme S1.

Round 2

Reviewer 3 Report

Comments and Suggestions for Authors

The authors have thoroughly and satisfactorily addressed all of this reviewer's comments.